Early prediction of student performance in CS1 programming courses

Llanos Jose jose.llanos@correounivalle.edu.co 1
Bucheli Víctor A. 1
Restrepo-Calle Felipe 2
1 School of Systems Engineering and Computing, Universidad del Valle Colombia , Cali , Valle del Cauca , Colombia
2 Department of Systems and Industrial Engineering, Universidad Nacional de Colombia , Bogotá , D.C. , Colombia
Wagner Stefan
Electronic publication date: 2023 Oct 31
Publication date: 2023
Volume: 9
Electronic Location ID: e1655
Received 2023 Apr 20; Accepted 2023 Sep 28
Copyright: ©2023 Llanos et al.
Copyright year: 2023
Copyright holder: Llanos et al.
License: This is an open access article distributed under the terms of the Creative Commons Attribution License, which permits unrestricted use, distribution, reproduction and adaptation in any medium and for any purpose provided that it is properly attributed. For attribution, the original author(s), title, publication source (PeerJ Computer Science) and either DOI or URL of the article must be cited.
License URL: https://creativecommons.org/licenses/by/4.0/

Keywords: Model prediction, Early prediction, Student performance, Predicting student performance, Programming course

Funding: The Corporación Universitaria del Huila—CORHUILA, and COLCIENCIAS sponsored the doctoral studies of Jose Llanos Mosquera This work was supported by the Corporación Universitaria del Huila—CORHUILA, and COLCIENCIAS sponsored the doctoral studies of Jose Llanos Mosquera. The funders had no role in study design, data collection and analysis, decision to publish, or preparation of the manuscript.

==============================
There is a high failure rate and low academic performance observed in programming courses. To address these issues, it is crucial to predict student performance at an early stage. This allows teachers to provide timely support and interventions to help students achieve their learning objectives. The prediction of student performance has gained significant attention, with researchers focusing on machine learning features and algorithms to improve predictions. This article proposes a model for predicting student performance in a 16-week CS1 programming course, specifically in weeks 3, 5, and 7. The model utilizes three key factors: grades, delivery time, and the number of attempts made by students in programming labs and an exam. Eight classification algorithms were employed to train and evaluate the model, with performance assessed using metrics such as accuracy, recall, F1 score, and AUC. In week 3, the gradient boosting classifier (GBC) achieved the best results with an F1 score of 86%, followed closely by the random forest classifier (RFC) with 83%. These findings demonstrate the potential of the proposed model in accurately predicting student performance.

Introduction

In programming courses, there is evidence of a high dropout rate, failure, and low academic performance, as reported in previous studies (Buenaño-Fernández, Gil & Luján-Mora, 2019; ElGamal, 2013). These issues have a negative impact on student engagement and their attitudes towards learning programming (Lu et al., 2018; Sivasakthi, 2017). To address these challenges, researchers have emphasized the importance of predicting student performance (Moreno-Marcos et al., 2020; Pereira et al., 2021). Performance in education refers to the degree to which students, teachers, or institutions achieve educational objectives, which is evaluated through continuous assessment of teaching effectiveness (Vilanova et al., 2019). In CS1 programming courses, various variables have been explored to understand their influence on student performance. The most significant variables include grades, study habits, and demographic factors.

Several research papers have focused on gathering data from programming courses to model and quantify factors related to student performance (Amra & Maghari, 2017; Ahadi, Lister & Vihavainen, 2016; Castro-Wunsch, Ahadi & Petersen, 2017; Costa et al., 2017; Estey & Coady, 2016; Leinonen et al., 2016; Pereira et al., 2019). Scholars argue that predicting performance is crucial as it enables teachers to identify at-risk students and provide early interventions (Aguiar & Pereira, 2018; Alamri et al., 2019; Costa et al., 2017; Hellas et al., 2018; Pereira et al., 2020b; Romero & Ventura, 2019). However, predicting student performance requires substantial efforts, ranging from student assessment to result processing.

To predict performance, Sunday et al. (2020) suggest utilizing variables that can influence grades. Pereira et al. (2020a) emphasize the importance of early-stage predictions to provide support to students in their learning journey. Kuehn et al. (2017) recommend making predictions and implementing corrective measures to ensure that at-risk students attain desired learning outcomes and grades.

Performance prediction is receiving increasing attention from researchers and educators, as it enables the analysis of student data and contributes to the educational process (Munson & Zitovsky, 2018; Pereira et al., 2020b; Romero & Ventura, 2019). The investigation of machine learning variables, characteristics, and algorithms that yield the best results in predefined metrics is currently underway (Hellas et al., 2018). In the context of programming courses, various studies have emerged to predict student performance early in the semester (Castro-Wunsch, Ahadi & Petersen, 2017; Dwan, Oliveira & Fernandes, 2017; Fwa, 2019; Pereira et al., 2019; Quille & Bergin, 2018; Romero & Ventura, 2019). This enables teachers to take targeted actions to assist students facing difficulties (Sun, Wu & Liu, 2020).

This article proposes a model for predicting student performance in weeks 3, 5, and 7 of a 16-week CS1 programming course. The model utilizes features such as grades, turnaround time, and number of attempts in three programming labs and an exam. The aim is to identify students with low and medium academic performance early on, who may be at risk of course failure or require intervention during the training process. The CRISP-DM methodology is employed, along with eight classification algorithms, and the model’s performance is evaluated using precision, recall, and F1 score metrics.

The structure of this article is as follows. The Related Work section presents the related work, including the characteristics, algorithms, and metrics used in predicting student performance in programming courses. The Methods section describes the CRISP-DM methodology. The Results section presents the experimental results. The Discussion and Conclusions section discusses the findings and provides conclusions. Finally, The Limitations of the Work section outlines the limitations of the research.

Related Work

Data sources used to predict student performance

The most commonly used data sources in predicting student performance are learning management systems or platforms (such as Moodle or edX) and automated source code evaluation tools like CodeWork and CodeBench, which allow for the collection of student-generated data and automated assessment of their submissions. Table 1 presents the type of data source, the specific tool used, the number of students, and the literature reference identified in the literature review, which are used to predict student performance in programming courses.

Table 1 Data sources used to predict student performance.

#	Data source	Tool	Number of students	Reference	
1	Learning management platform	edX	-	Moreno-Marcos et al. (2020)	
		Moodle	102	Lu et al. (2018)	
		Moodle	104	Xin & Singh (2021)	
		Moodle	336	Villagrá-Arnedo et al. (2017)	
		Moodle	32,593	Adnan et al. (2021)	
		Moodle	32,593	Waheed et al. (2020)	
		DARA	21,314	Sandoval et al. (2018)	
		Moodle	4,989	Conijn et al. (2016)	
2	Automated source code evaluation tools	CodeBench	2,058	Pereira et al. (2019)	
		CodeBench	2,056	Pereira et al. (2021)	
		CodeWork	86	Munson & Zitovsky (2018)	

Several articles related to student performance in programming courses were identified in learning management platforms. In Moreno-Marcos et al. (2020), two Java programming MOOCs hosted on edX were used. The first one was from Carlos III University of Madrid (10 weeks), and the second one was from Hong Kong University of Science and Technology (5 weeks). These courses involved programming tasks with peer review and automatically graded laboratories. Additionally, in Lu et al. (2018), data from a MOOC with 102 first-year students taking a Python programming course at a university in Taiwan were utilized. Tests were conducted with an experimental group of 48 students and a control group of 54 students. In Xin & Singh (2021), grades recorded in Moodle from 104 students enrolled in a distance learning course on Computer Engineering at the Open University of Madrid (UDIMA) were used for the development of a student attrition prevention system.

In Villagrá-Arnedo et al. (2017), data from the subject “Computational Logic” (336 students) in the Computer Engineering program were used. The data was collected from a website that stored information about the learning process and student-teacher interactions. Students downloaded exercise statements, uploaded their solutions, obtained exercise scores, and accessed their learning progress. Meanwhile, the teacher uploaded new exercise statements and monitored student progress.

In Adnan et al. (2021), a Learning Analytics dataset was used, which contained student information related to demographics and their interaction with the virtual learning environment (assessments, daily activities, number of clicks). The data was collected for seven courses and 32,593 registered students between the years 2013 and 2014. Similarly, in Waheed et al. (2020), the same dataset was used for the years 2014 and 2015 to analyze the resulting grades of students, categorizing them as distinction (3,024 records), pass (12,361 records), fail (7,052 records), and withdrawn (10,156 records).

In Sandoval et al. (2018), DARA is described as a system that stores demographic information of students at the time of enrollment, as well as their past and current academic status, including grades and enrolled courses. A total of 386,573 records were analyzed. This system was combined with the official SAKAY Learning Management System (LMS) of the university, which produces student activity logs. In total, information was collected from 21,314 university students over three semesters, from the second period of 2013 to the second period of 2014.

In Conijn et al. (2016), data on students’ online behavior was collected from different courses developed in Moodle and taught during the first semesters of 2014 and 2015 at Eindhoven University of Technology. The final sample included 4,989 students from 17 courses, with data covering eight weeks of course work and two weeks of final exams.

Regarding the automated source code evaluation tools used as data sources, the data related to student grading has been collected and utilized (Pereira et al., 2019). At the Federal University of Amazonas, data stored in the CodeBench tool from six introduction to programming courses were analyzed, involving a total of 2,058 students. This tool enables the automatic evaluation of exercises by validating the output of the code submitted by the student against the expected output provided by the instructor (Pereira et al., 2020a).

In Pereira et al. (2021), registration data from 2,056 students in a CS1 course (during the first two weeks) were used to track student progress and support blended learning methodology. Students solved programming problems using Python in assignments or exams, and a total of 150,314 code snippets were used for analysis.

In Munson & Zitovsky (2018), CodeWork, an automated evaluation tool, was utilized to gather student scores in programming tasks, exams, and the final course grades. A total of 61,684 compilation records from 86 students were analyzed.

Features used to predict student performance

In the literature review, several features that have been utilized for predicting student performance in programming courses were identified. This work defines four types of features: demographic data, grades, study habits, and programming. Demographic variables, such as age, gender, marital status, nationality, and income, have been employed in studies such as Costa et al. (2017), ElGamal (2013), Salinas, Williams & King (2019), Sivasakthi (2017) and Vilanova et al. (2019), as they are related to the student’s background. Additionally, other variables such as the number of credits passed, previous studies, and the semester have emerged in recent years as relevant factors in this type of research.

Although demographic features have been utilized in various research studies, Brooks, Thompson & Teasley (2015) states that gender, age, and race have limited predictive power for student performance. Furthermore, Alturki, Hulpus & Stuckenschmidt (2022) highlights that while demographic features are commonly used for performance prediction, their actual usefulness remains unclear. For instance, gender is often employed as a characteristic (Daud et al., 2017; Garg, 2018; Sultana, Khan & Abbas, 2017), yet some researchers have found that it does not significantly impact overall prediction accuracy (Ramesh, Parkavi & Ramar, 2013).

The grades category of features has been employed to consider various aspects, such as the student’s cumulative grades, grades from the previous year, and grades from the current semester, with the average grade point average (GPA) being the most commonly used metric (Quille & Bergin, 2019). Additionally, first-semester grades, completed assignments, lab grades, and final grades have also been utilized (Moreno-Marcos et al., 2020; Munson & Zitovsky, 2018; Pereira et al., 2019; Yoshino et al., 2020). In de la Peña et al. (2017), it is mentioned that this type of variable is significant for predicting student performance, as it enables the generation of tutorials and guidance throughout the academic process. Other authors indicate that grades serve as a key indicator of student performance, as they have yielded positive results in prediction models (Marbouti, Diefes-Dux & Madhavan, 2016; Zeineddine, Braendle & Farah, 2021).

Study habits represent another category of features utilized for predicting student performance in programming courses. These features encompass various aspects such as student forum engagement, clickstream data, class attendance, and video playback (Kuehn et al., 2017; Lu et al., 2018; Moreno-Marcos et al., 2018; Moreno-Marcos et al., 2020; Pereira et al., 2019; Sunday et al., 2020; Vilanova et al., 2019; Yoshino et al., 2020). They contribute to addressing issues of poor academic performance, low levels of student participation, and unsatisfactory grades obtained in the curriculum. Moreover, they aid in enhancing student aptitude for programming and fostering higher levels of engagement in collaborative programming courses (Abdulwahhab & Abdulwahab, 2017; Lu et al., 2018; Munson & Zitovsky, 2018; Salinas, Williams & King, 2019).

Finally, the programming category of variables has gained significant attention in recent years, enabling the analysis of factors that can influence programming learning (Lu et al., 2018; Sivasakthi, 2017). Within this category, variables related to mathematical background, problem-solving abilities, and prior programming experience have been explored (ElGamal, 2013). Additionally, the analysis of source code testing, executions, and keystrokes has provided valuable insights (Pereira et al., 2021). Furthermore, variables such as the number of attempts, average number of delivery attempts due to issues, number of accepted solutions, number of exercises completed, and results of submissions have been evaluated (Castro-Wunsch, Ahadi & Petersen, 2017; Costa et al., 2017; Leinonen et al., 2016; Moreno-Marcos et al., 2020; Munson & Zitovsky, 2018; Pereira et al., 2021; Pereira et al., 2020a; Pereira et al., 2019; Salinas, Williams & King, 2019; Sunday et al., 2020; Villagrá-Arnedo et al., 2017).

Table 2 illustrates the features and corresponding references of the research articles that were examined in the literature review and have contributed to the prediction of student performance in programming courses. The articles were categorized based on the types of features analyzed: demographic, grades, study habits, and scheduling.

Table 2 Features used in predicting student performance.

#	Feature type	Characteristics	Reference	
1	Demographic	Identification of the student, gender, age, marital status, previous studies, nationality, city, semester, number of credits passed, income.	Costa et al., 2017; ElGamal, 2013; Salinas, Williams & King, 2019; Sivasakthi, 2017; Vilanova et al. (2019)	
2	Grades	Intermediate grades of the tasks, final grade, grade of the first semester, completed assignment, laboratory work in class, qualification introductory programming test, final grade, exams of the period.	Buenaño-Fernández, Gil & Luján-Mora, 2019; de la Peña et al., 2017; Moreno-Marcos et al., 2020; Munson & Zitovsky, 2018; Pereira et al., 2020b; Sivasakthi, 2017; Sunday et al. (2020)	
3	Study habits	Participation in forums, clickstreams, class attendance, video playback, persistence in the development of activities, number of times you took a test, number of exam attempts per subject, numbers of logins.	Kuehn et al., 2017; Lu et al., 2018; Moreno-Marcos et al., 2018; Moreno-Marcos et al., 2020; Pereira et al., 2019; Sunday et al., 2020; Vilanova et al., 2019; Yoshino et al. (2020)	
4	Programming	Math background, problem-solving ability, previous programming experience, number of attempts, average attempts of submission by problems, number of accepted solutions, individual coding aptitude of the student, number of exercises performed, number of correct exercises, tests in the source code, results of submission, time to solve exercises, keystroke.	Castro-Wunsch, Ahadi & Petersen, 2017; Costa et al., 2017; Leinonen et al., 2016; Moreno-Marcos et al., 2020; Munson & Zitovsky, 2018; Pereira et al., 2021; Pereira et al., 2020a; Pereira et al., 2019; Salinas, Williams & King, 2019; Sunday et al., 2020; Villagrá-Arnedo et al. (2017)	

During the literature review, research articles were identified that predict student performance in the early stages of programming courses. The most commonly utilized features include demographics, grades, study habits, and programming behaviors. According to studies such as Alamri et al. (2019), Hellas et al. (2018), Pereira et al. (2020b) and Romero & Ventura (2019), these types of variables are crucial for predicting performance as they enable the identification of at-risk students. However, it is essential to employ automatic source code evaluation tools that facilitate the collection and analysis of variables related to attempts and delivery times in order to enable early student intervention (Pereira et al., 2020b).

Early performance prediction is receiving increasing attention, as it provides opportunities for improving educational outcomes through early interventions (Munson & Zitovsky, 2018; Romero & Ventura, 2019). Recent studies Castro-Wunsch, Ahadi & Petersen (2017), Dwan, Oliveira & Fernandes (2017), Fwa (2019) and Quille & Bergin (2018) propose methods to predict student performance in programming courses from the beginning, allowing teachers to take specific actions to assist students facing difficulties (Sun, Wu & Liu, 2020). Researchers in this field generally seek features that can be used in machine learning algorithms to generate more accurate predictions (Hellas et al., 2018).

Algorithms and metrics used in predicting student performance

Classification algorithms are machine learning techniques used to define features and classes based on different data. Their main objective is to classify new instances based on unknown data. In the process, features are used as input data, a model is trained using previously labeled examples, and a binary or multiclass class is predicted (Gama & Brazdil, 1995).

Regression algorithms are techniques used to predict numerical or continuous values based on input variables. In these types of algorithms, the relationships between independent (predictor) variables and the dependent (target) variable are analyzed, and a model is constructed to make predictions on new data. The goal of these algorithms is to find a mathematical function or relationship that best fits the training data. This function is then used to predict numerical values in new data instances (Massaron & Boschetti, 2016).

There are various classification and regression algorithms. Some examples of classification algorithms include k-nearest neighbors (KNN), decision tree (DT), support vector machine (SVM), and random forest (RF). Regression algorithms include linear regression, k-nearest neighbors regression (KNN regression), decision tree regression (DT regression), and random forest regression (RF regression). Each algorithm has its own characteristics, approaches, advantages, and disadvantages. However, the choice of algorithm depends on the specific problem and the type of available data (Singh, Thakur & Sharma, 2016).

This section discusses the machine learning algorithms and metrics commonly employed in predicting student performance in programming courses. The literature review revealed that classification algorithms are widely used due to their favorable predictive results. In recent years, the multilayer perceptron has gained prominence in certain research studies (Pereira et al., 2021; Pereira et al., 2020a). According to Buenaño-Fernández, Gil & Luján-Mora (2019), the outcomes obtained from these predictions demonstrate the effectiveness of machine learning techniques in forecasting student performance.

Classification algorithms have made significant contributions to research in this field. For instance, in Moreno-Marcos et al. (2020), these algorithms are utilized to predict student performance by considering intermediate assignment grades and the final course grade, with the aim of improving academic performance and reducing high dropout rates. Additionally, these algorithms have been combined with data mining techniques to assess and enhance student performance in programming courses (Sunday et al., 2020). Furthermore, they have been instrumental in predicting final grades based on historical grade performance, with the goal of reducing dropout rates and enhancing educational quality at universities (Buenaño-Fernández, Gil & Luján-Mora, 2019).

Other contributions of classification algorithms focus on predicting the academic performance of students in introductory programming courses, aiming to address poor performance (Sivasakthi, 2017). They have also been used to predict students’ performance based on weekly evaluations of programming exercises, indicating whether or not they will pass the course. This approach aims to tackle the high failure rate among programming students in the early stages (Yoshino et al., 2020). Similarly, in Pereira et al. (2019), classification algorithms were employed to predict the performance of programming students at the beginning of the course. Moreover, in Costa et al. (2017), this type of algorithm is used to predict early on which students will fail introductory programming courses, as a high rate of academic failure was identified.

In Pereira et al. (2021), classification algorithms are used to predict early behaviors that are related to the success or failure of programming students. This approach helps in generating effective interventions and improving ineffective behaviors that students exhibit while programming. On the other hand, in Abdulwahhab & Abdulwahab (2017), a model is developed to predict student behavior based on their grades. The aim is to support students who are achieving low grades in their coursework. In de la Peña et al. (2017), a real-time prediction model is created to identify whether a student is likely to drop out of a course based on historical grade data. The goal is to address the high dropout rate in programming courses.

Regression algorithms have made various contributions to the field of student performance prediction. For instance, prediction models have been developed to address poor academic performance, reduce high dropout rates, and mitigate the high failure rates observed among programming students. These models aim to enhance academic performance and enable teachers to provide timely interventions (de la Peña et al., 2017; Moreno-Marcos et al., 2020; Munson & Zitovsky, 2018; Pereira et al., 2019; Yoshino et al., 2020). Table 3 presents the algorithms, machine learning metrics, and references of the articles identified in the literature review that have been used in predicting student performance in programming courses.

Table 3 Algorithms and metrics used in predicting student performance.

#	Type of algorithm	Algorithms	Metric	Reference	
1	Classification	Naive Bayes (NB), Support Vector Machine (SVM), Support Vector Classification (SVC), Decision Trees (DT), K Neighbors Classifier (KNN), Multilayer Perceptron (MLP), and Random Forest (RF)	Area under the curve (AUC), validation cross, Spearman correlation, Pearson correlation, accuracy, precision, recall, and F1 score	Abdulwahhab & Abdulwahab (2017), Buenaño-Fernández, Gil & Luján-Mora (2019), Costa et al. (2017), Moreno-Marcos et al. (2020), Pereira et al. (2021), Pereira et al. (2019), Sivasakthi (2017), Sunday et al. (2020), Villagrá-Arnedo et al. (2017), Yoshino et al. (2020)	
2	Regression	Linear regression (LR), and logistic regression	Mean square error (RMSE), absolute error medium (MAE), R square (R2)	de la Peña et al. (2017), Moreno-Marcos et al. (2020), Munson & Zitovsky (2018), Pereira et al. (2019), Yoshino et al. (2020)	

The proposed model utilizes the classification algorithms listed in Table 3, as the literature review revealed that they achieved results above 75% in predicting student performance in the early weeks of the course. These algorithms have also incorporated features related to grades and programming, making them suitable for predicting performance in CS1 programming courses. Additionally, the model evaluates its performance using metrics such as accuracy, recall, and F1 score, which are employed in this research.

Early prediction in programming courses

Several research studies have contributed to early prediction in programming courses. For example, in Costa et al. (2017), a prediction model was developed to identify, starting from week 4, which students would drop out or continue in the course. The study utilized demographic data (age, gender, marital status, among others) and academic data (number of exercises completed by the student, performance in programming activities and weekly exams) from different introductory programming courses delivered both in-person and online. The classification algorithms used in the process included NB, DT, MPL, SVM, and the precision, recall, and F1 score metrics, achieving an 83% accuracy in the mentioned week.

Similarly, in Munson & Zitovsky (2018), a model was developed to predict student grades in the third week of the course. The data was collected from an automated source code evaluation tool called CodeWork, which stores students’ scores from programming activities (assignments and exams) as well as the final course grade. In the prediction, linear regression and the R2 metric were used, achieving an 87% accuracy. Other authors such as Pereira et al. (2019) predict whether a student will drop out or continue in the programming course starting from the second week. For data collection, they utilize CodeBench, an automated source code evaluation tool that stores student records related to programming activities, such as code testing, generated submissions, keystrokes, etc. In the process, they employed the time spent by the student (in minutes) for programming tasks, the DT classification algorithm, and the precision and recall metrics. In the conducted tests, the prediction model achieved 80% precision.

Table 4 presents the most relevant elements identified in related works where early prediction is generated in programming courses. These elements are related to the type of prediction and the week in which it is generated, the machine learning algorithms used in the process, the metrics employed, the prediction percentage, and the respective reference.

Table 4 Early prediction in programming courses.

What it predicts	Week prediction	Algorithms	Metrics	Prediction (%)	Reference	
Student drops out or not	4	NB, DT, MLP, SVM	Precision, Recall, F1 Score	83	Costa et al. (2017)	
Student grade	3	LR	R 2	87	Munson & Zitovsky (2018)	
Student drops out or not from the course	2	DT	Precision, Recall	80	Pereira et al. (2019)	
Student passes or fails the course	2	MLP, RF	Precision, Recall, F1 Score	82	Pereira et al. (2020a)	
The student passes or fails the course	2	XGBoost, MLP, RF	Precision, Recall, F1 Score	81	Pereira et al. (2021)	

In Pereira et al. (2020a), the prediction of whether a student passes or fails the introductory programming course is made in the second week. It utilizes 18 source code characteristics and programming behaviors stored in the CodeBench tool. The machine learning algorithms used are MLP, RF, and the metrics employed are recall, precision, and F1 score. The RF algorithm achieved the best results for the defined metrics, with an accuracy of 82%.

Furthermore, in Pereira et al. (2021), the prediction of whether a student passes or fails the CS1 course is made in the second week. The process involves using elements from the source code extracted from the CodeBench automated evaluation tool, such as tests, submissions, program executions, keystrokes, deadlines, error corrections, and time spent on programming. The classification algorithms used are XGBoost, MLP, and RF, along with metrics such as precision, recall, F1 score, and cross-validation of 10. They achieved an accuracy of 81% for the mentioned week.

Based on the literature review, the contributions generated by López Zambrano, Lara Torralbo & Romero Morales (2021), as well as the conferences organized by the International Educational Data Mining Society between 2020 and 2022, the following findings were identified:

• There are research works where learning management systems (Moodle, edX) are used as data sources to develop models for predicting student performance. Others include automated source code evaluation tools such as CodeWork and CodeBench to extract features for the proposed prediction models. However, in few studies, different data sources are combined, which may include static and dynamic records, to define relevant features in the construction of prediction models focused on student performance.

• The literature review identified four types of features for predicting student performance. However, in Ramesh, Parkavi & Ramar (2013), it is indicated that characteristics related to demographic data and study habits contribute little to these prediction models due to their static nature. On the other hand, Quille & Bergin (2019), Moreno-Marcos et al. (2020), Munson & Zitovsky (2018), Pereira et al. (2019), Yoshino et al. (2020) and López Zambrano, Lara Torralbo & Romero Morales (2021) states that features related to grades and programming are relevant for such models because they are dynamic in nature and have shown good results in predicting student performance.

• Several studies were also identified where features such as demographic data, prior studies, performance in activities, final course grades, clickstream data, forum participation, submissions, among others, were used. However, few research works include features such as delivery time and number of attempts, which, according to Castro-Wunsch, Ahadi & Petersen (2017) and Alamri et al. (2019), can be used for predicting student performance in programming courses.

• In early prediction of programming courses, it was identified that there are research studies that generate predictions between the 2nd and 4th week of the course. They employ classification algorithms such as naive Bayes (NB), decision tree (DT), multi-layer perceptron (MLP), support vector machine (SVM), random forest (RF), and XGBoost. Regression algorithms like linear regression (LR) are also used. The most commonly used metrics are precision, recall, and F1 score. The accuracy of the predictions can range from 80% to 87%. However, the use of multiple features in the process, as mentioned in Márquez-Vera, Morales & Soto (2013), leads to high dimensionality, which means a large number of features that can limit the prediction algorithms from achieving interesting results quickly. Additionally, it was observed that while most works utilize traditional machine learning algorithms to generate binary outputs, only a few studies combine traditional and ensemble algorithms to generate multiclass classifications.

Based on the findings described above, this research work proposes using the early prediction models from Costa et al. (2017), Pereira et al. (2020a) and Pereira et al. (2021) as a baseline and incorporating the contributions from Moreno-Marcos et al. (2020), Munson & Zitovsky (2018), Pereira et al. (2019), Castro-Wunsch, Ahadi & Petersen (2017), Alamri et al. (2019) and Márquez-Vera, Morales & Soto (2013) to define a multiclass classification model that predicts student performance using different data sources and incremental features related to grading (three labs) and programming (delivery time and number of attempts) in weeks 3, 5, and 7 of a 16-week CS1 programming course. Table 5 presents the best algorithm, the metric used to evaluate the model, the prediction percentage, and the references of the selected articles as the baseline.

Table 5 Baseline articles.

#	Best Algorithm	Metric	Prediction (%)	Reference	
1	SVM	F1 score	83	Costa et al. (2017)	
2	RF	F1 score	82	Pereira et al. (2020a)	
3	XGBoost	F1 score	81	Pereira et al. (2021)	

This work integrates two data sources into the proposed model. The first source comprises grades assigned to programming activities completed by students during the course. The second source is an automated source code evaluation tool that extracts students’ time spent on programming submissions and the number of attempts made. These dynamic records are then transformed into the model’s features. In contrast to other studies, our model predicts student performance early on with a limited number of variables and a reduced volume of available course data, achieving promising results through techniques such as feature selection and hyperparameter tuning, implemented via ensemble algorithms like GBC.

Methods

In this section, the research questions and the standard process for data mining (CRISP-DM) used in this work for the construction of the prediction model are described. The CRISP-DM lifecycle includes the following phases: data understanding, data preparation, modeling, and model evaluation.

In the understanding phase, the sources for data collection are defined, and the dataset is constructed. The objective is to observe the relationship between the records and verify the data quality. In the subsequent phase, the data is prepared using various data mining techniques to generate the required format for constructing the prediction model. In the modeling phase, algorithms and metrics are selected to implement in the model, and the prediction model is built. Finally, in the evaluation phase, the prediction model is assessed based on the defined metrics, and an estimation is made to determine if the proposed objectives have been achieved.

Research questions

Taking into account the problem described by Buenaño-Fernández, Gil & Luján-Mora (2019) regarding programming courses and their association with low academic performance, the following research questions are proposed:

• RQ1: How to build a model to early predict student performance in CS1 programming courses?

• RQ2: How to classify students in CS1 programming courses according to their performance?

Data understanding

Data sources

In this work, data was collected from three sources: (i) A .csv file containing the grades for laboratories 1, 2, and 3, exam 1, and the final grade of the student. These records correspond to 12 CS1 and CS2 programming courses conducted between the years 2021 and 2022. The assessed topics with achievement indicators were: inputs and outputs, decision structures, iterative structures, and functions. The grade values range from 0.0 to 5.0, and a student passes the course when the final grade is greater than or equal to 3.0. (ii) A .csv file extracted from an automated source code evaluation tool, including delivery time (in days), the number of attempts used by the student in their submissions, and the outcome achieved in the tool (did not submit, failed, overflow, success). (iii) A .csv file extracted from the academic records and admissions system of the University of Valle, Colombia, containing information for the period equal to 2022 or before 2022 and the enrollment type (withdrawn, regular, repeat) of the student during the semesters: 2021-1, 2021-2, and 2022-1.

Dataset

Considering the sources described in the previous section, the dataset was constructed for this work. The process involved integrating student grades, records from the automated source code evaluation tool, and data from the academic records and admissions system. Subsequently, a .csv file was created with 18 columns and 754 rows of the collected records. The columns include the course code, student name, grade for laboratory 1 with its corresponding delivery time, number of attempts, and the outcome achieved by the student according to the evaluation tool. Next, the grade for laboratory 2 appears along with the delivery time, number of attempts, and the outcome. Similarly, the grade, delivery time, number of attempts, and outcome for laboratory 3 are included. This is followed by the grade for exam 1, the academic period, the enrollment type, and the final grade of the student. Regarding the rows, it is important to clarify that each row corresponds to a student record. The programming activities and the features defined for constructing the prediction model are described below.

• Programming activities

Lab 1 comprises two programming exercises that are to be submitted in week 3 of the course. It carries a weight of 7.68% towards the student’s final grade. This lab assesses two achievement indicators: (1) implementation of an algorithm in a programming language that provides a solution to a problem. This includes input and output handling, decision structures, loops, arrays, and/or functions. (2) Application of a development methodology for a specific problem, including the delivery of elements for each stage of the methodology.

Lab 2 consists of two programming exercises that are completed in week 5 of the course. It contributes 12% towards the student’s final grade. The achievement indicators assessed in this lab are related to proposing algorithms that involve the use of decision structures and functions to solve a given problem.

Lab 3 and the exam both take place in week 7 of the course. The lab carries a weight of 6% towards the student’s final grade and assesses whether the student proposes an algorithm with repetition structures to solve a problem. The exam consists of two exercises and contributes 26.56% towards the final grade. It evaluates seven achievement indicators related to solving a problem that involves input and output, decision structures, cycles, arrays, and functions. It also assesses the student’s ability to perform desktop testing on algorithms that utilize decision structures, repetition structures, and functions. The exam further examines the student’s capability to propose algorithms that require the use of decision and repetition structures to solve a given problem and track recursive calls of a function.

• Features for constructing the prediction model

Each of the columns described in this section, with the exception of the course code, student name, and final grade, were defined as features for the proposed prediction model and were organized into three categories, along with their respective data type and description (see Table 6). The first category is related to the student’s grade in programming tasks, including the grades for the three laboratories and exam 1. The second category utilizes variables such as delivery time, number of attempts, and the outcome achieved by the student in the laboratories. The third category pertains to the academic period and enrollment type.

Table 6 Features used in the proposed model.

Category	Feature	Data type	Description	
Grade	lab_1	numeric decimal	Lab 1 grade	
	lab_2	numeric decimal	Lab 2 grade	
	lab_3	numeric decimal	Lab 3 grade	
	exam_1	numeric decimal	Exam 1 grade	
Automatic assessment tool	delivery_time_lab_1	numeric decimal	Lab 1 delivery time in hours	
	attempts_lab_1	integer numeric	Total lab 1 attempts	
	result_lab_1	integer numeric	Result delivered by INGInious for lab 1 (Did not file, Failed, Overflow, Success)	
	delivery_time_lab_2	numeric decimal	Lab 2 delivery time in hours	
	attempts_lab_2	integer numeric	Total lab 2 attempts	
	result_lab_2	integer numeric	Result delivered by INGInious for lab 2 (Did not file, Failed, Overflow, Success)	
	delivery_time_lab_3	numeric decimal	Lab 3 delivery time in hours	
	attempts_lab_3	integer numeric	Total lab 3 attempts	
	result_lab_3	integer numeric	Result delivered by INGInious for lab 3 (Did not file, Failed, Overflow, Success)	
Administrative information	period	integer numeric	Student enrollment period (Same as 2022, Less a 2022)	
	enrollment_type	integer numeric	Type of student enrollment (Retired, Normal, Repeater)	

Data preparation

Data anonymization

After constructing the dataset, the course code and student name columns were removed to anonymize the data. This was done to preserve student privacy, comply with data handling regulations, and foster trust in the use of personal data in research.

Data preprocessing

In this task, the resulting .csv file was loaded into a DataFrame to perform the necessary preprocessing. The first step was to identify the NaN records in the DataFrame and remove them from the dataset. The pandas function .isnull().sum(axis = 0) was used to obtain the number of NaN values for each column. Then, the .dropna() method was used to remove these records from the dataset, resulting in a total of 468 rows.

The next step involved transforming the outcome of the laboratories, academic period, and enrollment type into discrete values for classification. The sklearn library, specifically the preprocessing package and the LabelEncoder() method, were used for this task. The values for the laboratory outcome were encoded as follows: 0 for “Did not file,” 1 for “Failed,” 2 for “Overflow,” and 3 for “Success.” Similarly, the values for the academic period were encoded as 0 for “Same as 2022” and 1 for “Less than 2022.” Finally, for the enrollment type, the values were encoded as 0 for “Retired,” 1 for “Normal,” and 2 for “Repeater”.

The final task in data preprocessing involved transforming the student’s final grade column into the values proposed by Villagrá-Arnedo et al. (2017), which defines low, medium, and high performance. In our project, this is the target variable that we want to predict, and it was transformed as follows: 0 for Low Performance (grades between 0.0 and 2.9), 1 for Medium Performance (grades between 3.0 and 4.0), and 2 for High Performance (grades between 4.1 and 5.0).

Table 7 presents the number of students, laboratories, and exams submitted per semester after applying data preprocessing. A total of 3,744 submissions were analyzed across the three semesters. In the 2021-1 semester, there were 1,200 submissions from 150 students. In the 2021-2 semester, 1,400 submissions from 175 students were analyzed. Lastly, in the 2022-1 semester, there were 1144 submissions from 143 students.

Table 7 Number of students and deliveries per semester.

Semester	# Students	# Labs delivered	# Exams delivered	
2021-1	150	1,050	150	
2021-2	175	1,225	175	
2022-1	143	1,001	143	
Total	468	3,276	468	

Data balancing

In this task, the records of the target variable were grouped to observe the data distribution. For 0 (Low Performance), there were 162 records, while for 1 (Medium Performance) there were 200 records, and for 2 (High Performance) there were 106 records. Since the proposed prediction model is for classification, it was necessary to balance the data to prevent overfitting. Therefore, we used the oversampling technique, which includes the resample method from the sklearn library, to balance the records.

Modeling

Selected algorithms

The proposed model utilizes three types of classification algorithms: the traditional ones mentioned in Table 3, an artificial neural network called multilayer perceptron (MLP), and two ensemble algorithms: random forest (RF) and gradient boosting classifier (GBC). The traditional algorithms and MLP were selected because they have been used in previous research and have achieved good results (Buenaño-Fernández, Gil & Luján-Mora, 2019; Pereira et al., 2021; Pereira et al., 2020a). The last type of algorithms was selected because in the baseline articles (Pereira et al., 2020a; Pereira et al., 2021), the authors mention that they are suitable for predicting student performance in early stages as they improve performance and help stabilize the model.

After selecting the classification algorithms, the prediction model is built. For this, 80% of the data generated in the data preparation phase is used for training the model, and the remaining 20% is used for testing. Additionally, the grid search technique and cross-validation with a value of 10 were used to identify the best hyperparameters. Then, with the selected hyperparameters, the final prediction was generated, and the results for each of the selected metrics for weeks 3, 5, and 7 of the programming course were obtained. The model and data are available at https://zenodo.org/record/8111596.

Selected metrics

To evaluate the proposed prediction model in this work, the metrics identified in the literature review and baseline models were used, namely: precision, recall, and F1 score.

Precision is the rate of correctly identified or classified positive data. This metric indicates the true positive data within all instances classified by the machine learning algorithm (Ossa Giraldo & Jaramillo Marin, 2021). Its calculation is generated using the formula: (1) precision=TPTP+FP

Where:

• TP = True Positives

• FP = False Positives

Recall, also known as True Positive rate or Sensitivity, refers to the rate of correctly identified positive instances among all actual positive instances (Ossa Giraldo & Jaramillo Marin, 2021). Its equation is given by: (2) recall=TPTP+FN

Where:

• TP = True Positives

• FN = False Positives

F1 score is defined as the harmonic mean of precision and recall. This metric takes into account both false positives and false negatives (Sasaki, 2007). Its equation is given by: (3) F1=2Precision∗RecallPrecision+Recall

The value of F1 score is defined as the harmonic mean of precision and recall, thus its value lies between both metrics, leaning towards the lower value (Rodríguez Bustos, 2020).

Evaluation

Based on the baseline articles (Costa et al., 2017; Pereira et al., 2020a; Pereira et al., 2021), the objective for this work is to achieve a value greater than or equal to 83% in the F1 score metric, based on the prediction of student performance from week 3 of the course. All the results of the prediction model evaluation are presented in the results section.

Instruments

Two tools were utilized for conducting this research. The first tool is Replit (https://replit.com/), an online integrated development environment that enables the development of applications in various programming languages. Students utilize this platform to work on their course labs and exams, implementing solutions in a programming language. The second tool employed is INGInious (https://inginious.org/), an automated source code evaluation tool that assesses student submissions based on predefined input and output test cases provided by the instructor. Using INGInious, students upload their source code from Replit, containing their lab solutions. If the inputs and outputs are correct, a score of 100% is awarded. In cases where discrepancies are detected between the expected and submitted inputs/outputs, a comparison is displayed on the screen, allowing students to identify their errors and make subsequent submissions. Students have an unlimited number of attempts on INGInious, with the only constraint being the assignment’s due date.

Results

This section presents the results of the research questions defined in the methodology. The first part answers RQ1, based on the selection of features used in the construction of the proposed prediction model, the results achieved in the machine learning algorithms for the metrics Precision, Recall, F1 Score, and a radar chart that allows comparing the results achieved by each algorithm per week.

Features selected

Taking into account that the objective of this work is the early prediction of student performance, features related to the grade and automatic source code evaluation tool were used, specifically for Lab 1, as it is the first programming activity that students undertake in the course. These features were then combined with enrollment-related features for testing with the selected machine learning algorithms. However, the results obtained fell short of the objective set in this work. Therefore, the decision was made to remove some features from the prediction model, based on the findings described by Márquez-Vera, Morales & Soto (2013), which state that a large number of features can limit the results of classification algorithms in early prediction models.

To identify the features that have the greatest contribution to the prediction of the proposed model, the best features technique and the ELI5 library were used. The show_weights() method in ELI5 was employed to generate a report that includes the name of the feature and its weight contribution to the model, sorted in descending order. For the prediction of week 3, the selected features were lab_1, delivery_time_lab_1, and attempts_lab_1, which had weights of 0.48, 0.21, and 0.20, respectively. Subsequently, the prediction for week 5 was performed, combining the three types of features (grade, automatic evaluation tool, and enrollment) for Labs 1 and 2. This was done to create an incremental prediction model, which, according to de la Peña et al. (2017), improves the results for the selected metrics by adding new features. The show_weights() method from the ELI5 library was used again, and the selected features and their respective weights were: lab_1 = 0.35, lab_2 = 0.32, delivery_time_lab_1 = 0.13, and attempts_lab_1 = 0.10.

Finally, the prediction for week 7 was performed by combining the three types of features for Labs 1, 2, 3, and Exam 1. The selected features were lab_1, lab_2, lab_3, delivery_time_lab_1, and attempts_lab_1, based on the weights generated by the show_weights() method, which were 0.34, 0.31, 0.12, 0.08, and 0.06, respectively. The results described above indicate that the grade-related features contribute the most to the proposed prediction model. They have the highest weights in each of the combinations generated for weeks 3, 5, and 7.

Machine learning algorithms and metrics

In this work, eight classification algorithms were used for prediction, five of which are traditional algorithms: NB, SVC, DT, LR, and KNN, one is an artificial neural network: MLP, and the remaining two algorithms are ensemble algorithms: RF and GBC. The tests conducted allowed obtaining the values for the metrics: precision, recall, and F1 score for weeks 3, 5, and 7. In week 3, the algorithm that presented the highest results according to the selected metrics was GBC, with values of 0.86 for precision, 0.87 for recall, and 0.86 for F1 score. Meanwhile, the algorithm with the lowest results was NB, with 0.60, 0.63, and 0.61 for the same metrics.

In week 5, RF and GBC achieved 0.94 in all metrics, indicating that they are the algorithms with the best results. However, the NB algorithm obtained the lowest results with 0.67 for precision, 0.68 for recall, and 0.67 for F1 score. Finally, in week 7, it can be observed that GBC achieved the highest results, with 0.96 for each of the metrics. Meanwhile, the algorithm with the lowest results was NB with 0.76 for each of the metrics (see Table 8). Based on the results obtained, it can be observed that ensemble algorithms achieved the highest results for the selected metrics in weeks 3, 5, and 7. This indicates that this type of algorithms is appropriate for early prediction models, where characteristics related to grades, delivery time, and number of attempts generated by the student in programming activities are used.

Table 8 Results of prediction algorithms in week 3, 5 and 7.

Week	Metric	NB	SVC	DT	LR	KNN	MLP	RF	GBC	
3	Precision	0.60	0.75	0.78	0.65	0.80	0.77	0.84	0.86	
	Recall	0.63	0.75	0.78	0.67	0.81	0.78	0.84	0.87	
	F1 Score	0.61	0.75	0.77	0.65	0.79	0.77	0.83	0.86	
	AUC	0.83	0.87	0.92	0.85	0.94	0.89	0.95	0.96	
5	Precision	0.67	0.90	0.88	0.87	0.91	0.86	0.94	0.94	
	Recall	0.68	0.88	0.86	0.86	0.91	0.84	0.94	0.94	
	F1 Score	0.67	0.88	0.86	0.86	0.91	0.84	0.94	0.94	
	AUC	0.94	0.97	0.98	0.99	0.99	0.97	0.99	0.99	
7	Precision	0.76	0.95	0.91	0.91	0.93	0.90	0.95	0.96	
	Recall	0.76	0.94	0.91	0.90	0.93	0.89	0.95	0.96	
	F1 Score	0.76	0.94	0.91	0.90	0.93	0.89	0.95	0.96	
	AUC	0.96	0.99	0.99	0.99	0.99	0.99	0.99	0.99	

Figure 1 displays a radar chart illustrating the performance of the prediction algorithms in weeks 3, 5, and 7, based on the selected metrics. The lines closer to the center represent the results for week 3, where the algorithms achieved values ranging from 0.60 in accuracy to 0.87 in recall. The yellow, blue, and green lines located in the middle of the radar depict the results for week 5, with the algorithms reaching values between 0.67 for precision and F1 score, up to 0.94 for all metrics. Finally, the lines furthest from the center represent the results for week 7, with the algorithm range spanning from 0.76 to 0.96 for all metrics.

Figure 1 Radar graph with the results of the prediction algorithms per week.

Across the weeks, the algorithm with the highest performance was GBC, followed by RF in second place. Conversely, NB consistently achieved the lowest results across all three weeks. When comparing the performance of the algorithms week by week, it is evident that as the weeks progress and more features are incorporated into the prediction model, the results improve across all metrics for all algorithms.

Figure 2 displays the ROC curves of the GBC algorithm for the defined classes in the prediction model. The green line represents class 0, indicating low performance, with an area under the curve (AUC) of 0.90. The orange line represents class 1, representing medium performance, with an AUC of 0.64. The blue line represents class 2, corresponding to high performance, with an AUC of 0.89. The ROC curve provides insight into the sensitivity and specificity of the GBC algorithm at different threshold values. The GBC algorithm was selected as the best-performing algorithm based on the accuracy, recall, and F1 score metrics in weeks 3, 5, and 7.

Figure 2 Multiclass ROC curves for the GBC algorithm.

Classification of students according to their performance

After conducting tests on the model, we proceeded to predict the performance of a CS1 programming course for the semester 2022-2, in order to answer RQ2 based on statistical results obtained by the student. The course consists of 40 students, and the same metrics used in the tests (precision, recall, and F1 score) along with the ensemble algorithm GBC were employed. For the week 3, where the Lab 1 was developed, the model classified seven students as having low performance, two with medium performance, and 31 with high performance. In this activity, the student presents two programming exercises in INGInious, and two achievement indicators are evaluated. The first one is related to the use of a development methodology to solve a specific problem, and the second one involves implementing an algorithm in a programming language to solve a specific problem with inputs and outputs.

In Lab 2, which was developed in week 5, the prediction model classified five students as having low performance, eight with medium performance, and 27 with high performance. In this activity, two programming exercises were presented, and an achievement indicator was evaluated, which involved implementing decision structures and functions to solve a specific problem.

Finally, for week 7 where Lab 3 was conducted, the model classified seven students as having low performance, eight with medium performance, and 25 with high performance. In this learning activity, two programming exercises were developed, and an achievement indicator related to the implementation of an algorithm with repetition structures to solve a specific problem was evaluated. This information is important because the instructor can make decisions regarding the course and intervene with students who exhibit low or medium performance as early as week 3 of the course. This contributes to addressing the issue of low academic performance that is often encountered in CS1 programming courses.

Statistical results of students’ performance

Then, the mean and standard deviation of the students classified as “Low”, “Medium”, and “High” performance were analyzed for each of the characteristics defined in the prediction model (grade, delivery time, and number of attempts). The objective of this analysis is to observe the distribution of the data so that the professor can identify student behaviors and provide support throughout the academic semester (see Table 9).

Table 9 Results by activity and characteristic for students classified as “Low Performance” (LP), “Medium Performance” (MP), and “High Performance” (HP) are as follows.

Week	Activity	Feature	LP (mean)	LP (SD)	MP (mean)	MP (SD)	HP (mean)	HP (SD)	Kruskal Wallis (p-value)	
3	Lab 1	Grade	0.00	0.00	2.90	0.56	4.90	0.24	2.9e−07	
		Delivery time	0.00	0.00	1.81	1.67	2.22	2.11	4.7e−05	
		Attempts	0.00	0.00	2.63	2.06	4.51	3.18	5.9e−05	
5	Lab 2	Grade	1.00	2.23	3.60	1.23	4.70	0.47	4.2e−04	
		Delivery time	0.01	0.03	1.10	2.64	0.05	0.03	1.1e−03	
		Attempts	3.50	0.70	2.43	1.96	2.13	1.56	0.052	
7	Lab 3	Grade	0.35	0.94	1.88	1.63	4.80	0.69	2.2e−0.7	
		Delivery time	1.07	2.63	4.43	3.46	5.63	2.49	0.064	
		Attempts	3.00	4.12	2.91	2.53	2.55	2.40	0.080	

In the statistical analysis of the data, three important findings were identified that can contribute to the performance of students in CS1 programming courses. The first is related to the standard deviation of grades, the second is related to the mean time used by students in programming assignments, and the last is related to the mean number of attempts.

In the standard deviation of grades, it is observed that the average for students with low performance is 1.06, while those with medium performance scored 1.14 and those with high performance obtained 0.47. These results indicate that students with high performance achieved the lowest dispersion in grades, indicating that the values tend to cluster closer to the mean. On the other hand, students with medium performance had the highest standard deviation, showing that the values tend to spread more widely around the mean. This statistical measure is useful because it allows the teacher to understand the variability of grades generated by students.

For the delivery time used by students in programming activities, the mean for students with low performance was 0.36 days, medium performance was 2.45 days, and high performance was 2.63 days. Comparing these results, it can be observed that students with low performance used less time in delivering programming labs, while students with high performance used 0.18 days more than students with medium performance and 2.27 days more than those with low performance. These results are likely to have an influence on students’ academic performance, but further tests are needed to validate this claim.

In the number of attempts used by students in programming submissions, the mean for students with low performance was 2.17, for medium performance it was 2.66, and for high performance it was 3.06. These results indicate that students with high performance make more submissions in programming labs compared to students with medium and low performance. This information can be relevant for academic performance because students with high performance tend to be more persistent with their submissions and aim to achieve higher grades in the automated evaluation tool.

Finally, the non-parametric statistical test of Kruskal-Wallis was applied. This test is used to determine if there are significant differences at a statistical level between two or more groups. The test determines if the medians of two or more groups are different by calculating a test statistic and comparing it to a cutoff point in the distribution. In this work, three groups were defined for the analysis: the first group comprised students classified as “Low Performance”, the second group comprised students classified as “Medium Performance”, and the last group comprised students classified as “High Performance”. Two hypotheses were defined: H0 states that the medians of the groups are equal, and H1 states that the medians of the groups are not equal. In this sense, if the resulting p-value is less than 0.05, H0 is rejected. For the tests conducted in this work, as shown in Table 9, it was observed that the p-value is less than 0.05 for the grade of all weeks, the delivery time of the labs 1 and 2, and the number of attempts for week 3. Therefore, H0 is rejected, and it is concluded that for the grades, delivery time, and number of attempts described above, the medians of the groups are not equal.

DISCUSSION AND CONCLUSIONS

In the literature review, it was identified that the most commonly used data sources for predicting student performance are learning management platforms (such as Moodle or edX) and automatic source code evaluation tools like CodeWork and CodeBench. However, few studies combine different data sources. In this work, three different data sources are used to construct the prediction model dataset. These sources include student grades, records from an automatic source code evaluation tool, and enrollment data from an academic records and admissions system.

Demographic characteristics and study habits are commonly used to predict student performance. However, Ramesh, Parkavi & Ramar (2013) suggests that these types of characteristics contribute little to early prediction models due to their static nature. Some authors, such as Quille & Bergin (2019), Moreno-Marcos et al. (2020), Munson & Zitovsky (2018), Pereira et al. (2019), Yoshino et al. (2020) and López Zambrano, Lara Torralbo & Romero Morales (2021), recommend using grading and programming characteristics for predicting student performance as they have achieved good results. In this work, we utilized these two types of characteristics and validated the findings described by these authors, as we achieved results exceeding 83% for the F1 Score metric in week 3 of a CS1 programming course, as defined in the methodology.

Various research studies utilize demographic data, prior studies, activity grades, final course grades, clickstream data, forum participation, and submissions as characteristics to predict student performance. However, few studies include the characteristics of delivery time and number of attempts. According to Castro-Wunsch, Ahadi & Petersen (2017) and Alamri et al. (2019), these characteristics can be used for predicting student performance in programming courses. In the proposed model, we used these characteristics (delivery time and number of attempts) and combined them with grades to predict student performance in programming courses at an early stage. In the tests conducted, the GBC algorithm achieved an 86% F1 Score metric for week 3 with 468 records. This result confirms that the mentioned characteristics can be used for this type of prediction. Additionally, we validated the findings described in Pereira et al. (2020a); Pereira et al. (2021), where it is mentioned that ensemble algorithms can yield better results than traditional algorithms. This is because the algorithm is based on decision trees that are repeatedly summed, and each iteration corrects the errors found in the leaves of the previous tree using statistical measures. For this reason, ensemble algorithms are suitable for predicting student performance at early stages.

With the development of this work, we also validated the findings described by Márquez-Vera, Morales & Soto (2013) in their research, where they indicate that early prediction models can be created with few characteristics. In this case, for week 3, we used the grade of Lab 1, delivery time, and the number of attempts generated by the student in the same programming activity, and achieved results above 85% for the precision, recall, and F1 score metrics. Additionally, we validate the findings described by de la Peña et al. (2017) in their work, where it is mentioned that the metrics improve for prediction models when they are incremental and new characteristics are added to the prediction model. In the results of the proposed model, it can be observed that when transitioning from week 3 to week 5 and adding new characteristics, the model reaches 94% in the F1 score metric for the RF and GBC algorithms. Furthermore, when transitioning from week 5 to week 7, the GBC algorithm can reach 96% by adding new characteristics for the same metric.

In the executed tests, it was observed that the characteristics related to grades proved to be the most important for the proposed prediction model, as they obtained higher weights compared to the characteristics related to the automatic source code evaluation tool and enrollment. It was also observed that as we move from one week to another and add new characteristics, the results of the metrics change for all algorithms, increasing approximately 7% from week 3 to week 5, and 6% from week 5 to week 7.

To classify the students in the programming course according to their performance, the mean, standard deviation, and Kruskal-Wallis test were used. In the statistical analysis, it was identified that students with low and medium performance have higher dispersion compared to students with high performance, with values of 0.59 and 0.67, respectively. It was also observed that students with high performance take more time in their programming lab submissions compared to students with low and medium performance, with a difference of up to 2.27 days. In terms of the number of attempts made by students in their programming submissions, it was found that students with high performance make more submissions in the labs compared to students with low and medium performance, with values of 0.89 and 0.40 attempts, respectively. These findings are important for the academic performance of students, as the instructor can understand the variability of grades generated by students, make informed decisions for course development, and intervene early with students who may be facing difficulties.

As a future work, it is suggested to develop a tool that integrates the proposed prediction model in order to provide early intervention to students in weeks 3, 5, and 7 of CS1 programming courses. This tool would enable instructors to monitor the academic performance of the course, make informed decisions regarding the educational process, and provide support to students facing difficulties.

Limitations of the Research

This research has several limitations, and it is important to be aware of them in order to properly evaluate the results. This work is limited to predicting student performance in weeks 3, 5, and 7 of a 16-week CS1 programming course. The dataset used for training and testing the prediction model consists of 468 records from 12 CS1 and CS2 programming courses conducted in 2021 and 2022. The selected features are limited to grade, delivery time, and number of attempts generated by students in three programming labs within the course. The selected algorithms are limited to NB, SVC, DT, LR, KNN, MLP, RF, and GBC, with the metrics precision, recall, and F1 score.

Supplemental Information

Supplemental Information 1 Model

Click here for additional data file.

Supplemental Information 2 Data

Click here for additional data file.

Additional Information and Declarations

Competing Interests

Author Contributions

Data Availability

The authors declare that there are no competing interests.

Jose Llanos conceived and designed the experiments, performed the experiments, analyzed the data, performed the computation work, prepared figures and/or tables, authored or reviewed drafts of the article, and approved the final draft.

Víctor A. Bucheli conceived and designed the experiments, prepared figures and/or tables, authored or reviewed drafts of the article, and approved the final draft.

Felipe Restrepo-Calle conceived and designed the experiments, prepared figures and/or tables, authored or reviewed drafts of the article, and approved the final draft.

The following information was supplied regarding data availability:

The data and model are available in the Supplemental Files and at Zenodo: Jose Llanos, Victor Bucheli, & Felipe Restrepo. (2023). Early prediction of student performance in CS1 programming courses. https://doi.org/10.5281/zenodo.8111596.

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
