# Peer review of "Early prediction of student performance in CS1 programming courses"

_PeerJ Computer Science, doi:10.7717/peerj-cs.1655_

## Round 0.1 · original submission · Major Revisions

While I not necessarily agree with Reviewer 1 that the paper needs a new perspective, I do agree that you need to make more clear how you contribute to the body of knowledge on early predictions. Please make fully clear what is missing when discussing the related work and how you contribute there.

Furthermore, the methodology needs more details. For example, the design decisions and implications when building the model need to be discussed in much more detail. Also the evaluation should build on existing work and make clear comparisons. Why don't you use the best approaches as baseline?

Please do not rely only on GitHub but move the data and model to a permanent archive such as Zenodo.

It is essential to clearly and in detail discuss all the limitations of the approach and the analysis. Right now, it is far too short.

Reviewer 1 ·

Basic reporting

Authors perform an analysis of early prediction of student performance in programming courses. The article is well written and it has a very good structure and it is easy to follow and understand. I also like the way related work is organized as it addresses different important issues related to prediction and it covers enough literature. In addition, the raw data is shared and the quality of figures and tables is good. The objectives are also clear and the paper is technically sound.

Experimental design

However, I am concerned about the originality of the paper because although the experimental design and methodology is fine, there are many similar papers addressing the same issue and the temporal analysis to discover how much student performance can be anticipated has been largely analyzed. While more research can always be done, I believe it should show a different perspective or add different analyses to make the paper relevant for the journal (e.g., more research questions could be added to explore more things).

Validity of the findings

As you mention in the literature, other papers have found similar results. I appreciate your discussion section because you present other similar articles but I expect a clearer and significant contribution for a journal paper.

Additional comments

Apart from the previous concerns, I have a couple of minor comments: in section 0.2, you should not only mention “classification algorithms” but specify which algorithms are. Moreover, you should justify your metrics in 0.6 and I would recommend providing AUC in the results of Table 6.

Cite this review as

·

Basic reporting

The paper addresses a relevant problem in educational data mining, considering the research question on how to predict early student performance in CS1 programming courses to identify who is at risk of losing it. Hence, the overall goal of the work presented in the is to explore supervised machine learning techniques to understand their performance in terms of how to predict, as early as possible, students’ outcomes in programming courses. The idea is to make earlier predictions about which students performance, using only the most important attributes. The main research question is well stated. The obtained results with the used techniques are, somewhat, compared. These results might be useful for the university where the investigation has been conducted. However they are not so novel for the educational data mining community.

Overall, the paper is well-organized, but it is not so clear. In addition, many ideas are introduced but they are not explained in detail or in a way generally organized manner. Furthermore, the authors should ask a native English speaker to read through the entire paper, as there are quite a few grammatical errors. Although the goal of the paper is interesting it is not clear what are the main differences between the proposed approach and others defined by the education data mining community.

Experimental design

The main research question is well stated, but in my opinion it should be broken into more specific research questions.
The methodology used in the cases presented could benefit from additional detail, and the study's limitations should be more clearly acknowledged.

Validity of the findings

In my opinion, the proposed approach has little originality in relation to some works already published, as for example the papers of the researcher Cristóbal Romero and his team. The problem of early prediction is widely explored in data mining approaches, it is not clear what is missing in the existing works. There is much previous work addressing feature selection method and early prediction in Education Data Mining to predict students’ academic performance, including a technical and comparative perspective on the performance of the algorithms.

While the paper presents some potential interesting findings, there are several limitations that should be addressed. The methodology used in the cases presented could benefit from additional detail, and the study's limitations should be more clearly acknowledged. The major claims and conclusions of the paper are not well substantiated. The findings seems to be important, but they are presented with a limited significance.

Additional comments

While the paper presents some potential interesting findings, there are several limitations that should be addressed. The paper should describe and justify its main technical choices. The methodology used in the cases presented could benefit from additional detail, and the study's limitations should be more clearly acknowledged.

In my opinion, the proposed approach has little originality in relation to some works already published, as for example the papers of the researcher Cristóbal Romero and his team. The problem of early prediction is widely explored in data mining approaches, it is not clear what is missing in the existing works, including the ones mentioned in this paper. There is much previous work addressing feature selection method and early prediction in Education Data Mining to predict students’ academic performance, including a technical and comparative perspective on the performance of the algorithms.

The major claims and conclusions of the paper are not well substantiated. The findings seems to be important, but they are presented with a limited significance. I recommend you to improve this aspect in the text.

Cite this review as

---

## Round 0.2 · Minor Revisions

The reviewer and I am happy with the improvements of the paper. Please include the concluding paragraph as suggested by the reviewer to summarise the novelty of the paper. I do not see any other problems with the paper.

Reviewer 1 ·

Basic reporting

Authors have properly addressed my comments

Experimental design

While authors have include more details about the literature review in 0.1 and this is fine, the section finishes with the details of a reference. I would expect a concluding paragraph that serves to justify the novelty of the paper.

Validity of the findings

Authors have properly addressed my comments

Additional comments

Authors have properly addressed my comments

Cite this review as

---

## Round 0.3 · accepted · Accept

Thanks for addressing all remaining concerns of the reviewers. I looked through the submission myself again and strongly support the publication of your manuscript.